# *Pneumocystis* spp. in Pigs: A Longitudinal Quantitative Study and Co-Infection Assessment in Austrian Farms

**DOI:** 10.3390/jof8010043

**Published:** 2021-12-31

**Authors:** Barbara Blasi, Wolfgang Sipos, Christian Knecht, Sophie Dürlinger, Liang Ma, Ousmane H. Cissé, Nora Nedorost, Julia Matt, Herbert Weissenböck, Christiane Weissenbacher-Lang

**Affiliations:** 1Department for Pathobiology, Institute of Pathology, University of Veterinary Medicine Vienna, Veterinärplatz 1, 1210 Vienna, Austria; Barbara.Blasi@vetmeduni.ac.at (B.B.); Nora.Nedorost@vetmeduni.ac.at (N.N.); Julia.Matt@vetmeduni.ac.at (J.M.); Herbert.Weissenboeck@vetmeduni.ac.at (H.W.); 2Department for Farm Animals and Veterinary Public Health, Clinic for Swine, University of Veterinary Medicine Vienna, Veterinärplatz 1, 1210 Vienna, Austria; Wolfgang.Sipos@vetmeduni.ac.at (W.S.); Christian.Knecht@vetmeduni.ac.at (C.K.); Sophie.Duerlinger@vetmeduni.ac.at (S.D.); 3Critical Care Medicine Department, NIH Clinical Center, National Institutes of Health (NIH), 10 Center Drive, Bethesda, MD 20892, USA; mal3@cc.nih.gov (L.M.); ousmane.cisse@nih.gov (O.H.C.)

**Keywords:** *Pneumocystis* spp., pig, farms, respiratory diseases, PRDC

## Abstract

While *Pneumocystis* has been recognized as both a ubiquitous commensal fungus in immunocompetent mammalian hosts and a major opportunistic pathogen in humans responsible for severe pneumonias in immunocompromised patients, in pigs its epidemiology and association with pulmonary diseases have been rarely reported. Nevertheless, the fungus can be quite abundant in porcine populations with up to 51% of prevalence reported so far. The current study was undertaken to longitudinally quantify *Pneumocystis carinii* f. sp. *suis* and other pulmonary pathogens in a cohort of 50 pigs from five Austrian farms (i.e., 10 pigs per farm) with a history of respiratory disease at five time points between the first week and the fourth month of life. The fungus was present as early as the suckling period (16% and 26% of the animals in the first and the third week, respectively), yet not in a high amount. Over time, both the organism load (highest 4.4 × 10^5^ copies/mL) and prevalence (up to 88% of positive animals in the third month) increased in each farm. The relative prevalence of various coinfection patterns was significantly different over time. The current study unravelled a complex co-infection history involving *Pneumocystis* and other pulmonary pathogens in pigs, suggesting a relevant role of the fungus in the respiratory disease scenario of this host.

## 1. Introduction

The genus *Pneumocystis* comprises fungal species of opportunistic respiratory pathogens that are responsible for severe and potentially lethal pneumonia in immunocompromised humans and other mammals. The fungus was first observed in 1909–1910 by C. Chagas and A. Carini in guinea pig and rat lungs, respectively, and in both cases falsely classified as *Trypanosoma* spp. [1]. Shortly thereafter, the Delanoës [2] recognized it as a new species and named it *Pneumocystis* (*P.*) *carinii*. It was not until molecular approaches became available in the late 1990s that it was possible to firmly establish that *Pneumocystis* belonged to the fungal kingdom and that those infecting different hosts are distinct species [3]. Recent comparative genome studies of *P. jirovecii*, *P. carinii*, and *P. murina* further confirmed this classification [4]. While the majority of studies on this fungus have focused on humans and laboratory animals, *Pneumocystis* has also been reported in various other mammals, including wild and domesticated farm and pet animals [5,6,7], often with very low organism loads. From an epidemiological point of view, little is known so far about *Pneumocystis* in pigs, previously named as *Pneumocystis carinii* f. sp. *suis* [8] and referred as *P. suis* hereafter in the current study. The first reports of *Pneumocystis* in pigs dated back to the late 50s and 60s, when Bondy observed it in a group of piglets and later on Nikolskij and Kucera in a piglet and an elderly pig as well [9,10,11]. In the porcine host, this fungus is associated with mild lung lesions, mostly as interstitial pneumonia, even though granulomatous inflammation can also be observed [12]. Its presence has been successively reported in studies conducted on swine farms or abattoirs in Austria, Denmark, Portugal, Brazil, Japan, and South Korea [13,14,15,16,17,18,19,20]. Its prevalence ranges between 7 and 51% [15,17,21], with variation according to geographic distribution, climate, and farm management [22]. This variation was also observed in our annual routine diagnostics data (Blasi, unpublished data). It is currently not completely clear whether the fungus causes a subclinical infection or is rather a true pulmonary pathogen in pigs, as a *P. suis* single-infection has been rarely reported, especially in relation to specific clinical symptoms. Indeed, with regard to respiratory diseases, the pig typically presents a poly-microbial and multifactorial scenario, which is commonly described with the term Porcine Respiratory Disease Complex (PRDC). Both primary and secondary pathogens are involved, presumably with the first paving the way for the second, by either causing local tissue damage and/or impairing the host’s immune system. In general, respiratory diseases in pigs are among the most relevant concerns for swine producers due to the huge economic losses associated with increased treatment costs, reduced growth rates, and increased mortalities [23,24]. For example, Holtkamp et al. reported a yearly loss in the USA swine industry of over $600 million owing only to the Porcine Reproductive and Respiratory Syndrome (PRRS) [25]. Due to the high number of microorganisms involved, in depth understanding of the complex synergistic effects of various specific co-infections and environmental factors that play a role in PRDC is still a critical goal to achieve.

Diagnostic studies in *P. suis*-positive specimens consistently revealed co-infections with other pulmonary pathogens including Porcine Circovirus type 2 (PCV2), PRRSV, *Pasteurella multocida*, *Streptococcus suis*, and *Mycoplasma hyopneumoniae* [14,26]. However, given the retrospective or cross-sectional nature of those studies it was neither possible to assess interactions between *P. suis* and the co-infecting agents nor to evaluate their age-dependent incidence. The current study was thus undertaken to determine the dynamics of a *P. suis* infection and coincident pulmonary co-infections by using a longitudinal experimental design. We sampled bronchoalveolar lavage fluid (BALF) and serum of pigs from five different farms at five time points and quantified *P. suis* and other pulmonary pathogens including PCV2, PRRSV, Swine Influenza Virus (SIV), *Actinobacillus pleuropneumoniae* (*A.p.*), *Bordetella bronchiseptica* (*B.b.*), *Bordetella pertussis* (*B.pe.*), *Bordetella parapertussis* (*B.pa.*), *Mycoplasma hyopneumoniae* (*M.hp*.), *Mycoplasma hyorhinis* (*M.hr*.), *Mycoplasma flocculare* (*M.fl*), *Pasteurella multocida* (*P.m.*), *Glaesserella parasuis* (*G.p*.), and *Streptococcus suis* (*S.s.*). Moreover, clinical symptoms of each pig were also documented at every time point and analyzed for any correlations with the types and quantities of pulmonary pathogens detected.

## 2. Materials and Methods

### 2.1. Sample Collection and Nucleic Acid Isolation

BALF and serum samples were collected from a group of ten pigs (Large White ×Landrace) each in five geographically distant conventional farms located in Lower and Upper Austria between September 2019 and January 2020. The sampling procedures were in accordance to Animal Trial legislation and approved by the national authorities on 8 April 2019 (GZ 68.205/0063-V/3b/2019). All farms had documented records of swine respiratory diseases. Four out of five were farrow-to-finish farms while one farm (Farm 5) was a piglet producer. Farms 1, 2, and 4 were one-site farms, while Farm 3 was a multi-site farm. The size of the farms ranged from 110 to 200 sows and from 300 to 900 fatteners. For sample collection, piglets exhibiting clinical signs of respiratory disease were selected and anaesthetized with a combination of 2 mg/kg of azaperone and 10 mg/kg of ketamine administered intramuscularly before sampling. For each pig, sterile collection equipment was used and the samples were handled cautiously and processed under sterile conditions to prevent sample-to-sample cross-contamination. The pigs were placed in sternal recumbency. A flexible tube was inserted into the trachea using a laryngoscope and then fixed against the hard palate. Sodium chloride solution (approx. 20 mL) was then instilled and recollected by aspiration. Each pig was sampled at five consecutive time points: first and third week, and second, third, and fourth month of life (1-WO, 3-WO, 2-MO, 3-MO, 4-MO, respectively). DNA and RNA were extracted from a total of 248 BALF samples (only 2 pigs could not be sampled successfully) using the MasterPure Complete DNA and RNA Purification Kit (Epicentre, Biozym, Vienna, Austria) and 250 serum samples using the QIAamp Viral RNA Mini Kit (Qiagen, Vienna, Austria). Both kits allow for the simultaneous extraction of RNA and DNA from different specimens. DNA and RNA were quantified with a DS-11 Spectrophotometer (DeNovix, Wilmington, DE, USA).

### 2.2. Quantitative Detection of Pulmonary Pathogens by Real Time PCR

Real time quantitative PCR (qPCR) was applied on single-copy genes in order to detect and quantify *P. suis*, PCV2, and bacterial agents. Reverse transcription quantitative PCR (RT-qPCR) was used to detect PRRSV-EU, PRRSV-US, PRRSV-HP, and SIV. The investigated pathogen species and the respective target genes along with PCR primers and probes are listed in Table 1. Two technical replicates were run for each sample on the 7500 Fast Real-Time PCR System (Applied Biosystems, Waltham, MA, USA). We used the Luna Universal Probe qPCR Master Mix (New England Biolabs, Frankfurt, Germany) for qPCR at the following conditions: 60 s at 95 °C, 40 cycles of 15 s at 95 °C, and 30 s at the target specific temperature (Table 1). Given the current knowledge regarding the existence of two distinct *P. suis* clusters in mitochondrial large subunit (mtLSU) and small subunit (mtSSU) rRNA sequences [27], we targeted the single-copy gene *sodA* in its two variants (*sodA*1 and *sodA*2) for the quantification of *P. suis* in this study. The Luna Universal Probe One-Step RT-qPCR Kit (New England Biolabs, Frankfurt, Germany) was used for RT-qPCR at the following conditions: reverse transcription step for 10 min at 55 °C, initial denaturation for 60 s at 95 °C, and 40 cycles of 15 s at 95 °C and 30 s at the target specific temperature (Table 1). PCV2 and PRRSV-EU, -US, and -HP strains were quantified from serum while for all other pathogens this was done using BALF. In case of *Bordetella* spp., the Bordetella Real TM Kit (Sacace Biotechnologies, Como, Italy) was used. The kit contains a DNA-based internal control to monitor any potential inhibition of the qPCR reaction. To assess any potential inhibition of the RT-qPCR reactions for virus detection, we established RT-qPCRs for the host β-actin (ACTB) and β2-microglobulin (B2M) genes. We used the following primers and probe for the porcine β-actin gene: ACTB-F 5′-AAAGACCAGAAACCAAGCG-3′, ACTB-R 5′-CCACAAAGACCAGACACAAG-3′, ACTB-Probe 5′-CCAGACGCCAGATGACAGTCACCTGGGCA-3′ (annealing temperature 60 °C) and the following for the B2M gene: B2M-F 5′-TTCAGGTTTACTCACGCCAC-3′, B2M-R 5′-TTAGTGGTCTCGATCCCA-3′ and B2M-Probe 5′-TGGTCTTTCTACCTTCTGGTCCACACTGAGTT-3′ (annealing temperature 58 °C, primers were modified from Le et al. [28]). None of the BALF or serum samples tested showed any inhibition.

The amplification products of each qPCR were cloned in by using the TOPO TA Cloning Kit (Invitrogen, Life Technologies, Vienna, Austria) with a pCR^TM^ 2.1 TOPO vector and TOP10 chemically competent cells. Ten separate colonies were selected from the Luria-Bertani (LB) agar (Invitrogen, Life Technologies, Vienna, Austria) and incubated over night at 37 °C in LB broth (Invitrogen, Life Technologies, Vienna, Austria). Plasmid DNA was extracted using the QIAprep Spin Miniprep Kit (Qiagen, Vienna, Austria) and the concentration was determined with a DS-11 Spectrophotometer (DeNovix, Wilmington, DE, USA). For the RNA-based viruses SIV and PRRSV (-EU, -US, and -HP), the cloned DNA was used as a template for the T7 polymerase Ampliscribe^TM^ from the T7-Flash Transcription kit (Lucigen, Roche, Vienna, Austria). The obtained RNA was quantified with a DS-11 Spectrophotometer (DeNovix, Wilmington, DE, USA). A 10-fold dilution of the targeted DNA or RNA was prepared for the determination of the standard curves for all target sequences in the qPCR and RT-qPCR, respectively. 

### 2.3. G. parasuis Serotyping and SIV Subtyping

A subset of *G.p.* positive samples was analysed for their serovars. In particular, a PCR assay targeting the *ompP2* gene was established using primers 5′-GCAGCATCAGCATCAGCTGTAACART-3′ and 5′-CCAACACCAAGTGCTTKGTCAGTAAC-3′. PCR was performed in a 25 µL total reaction containing 12.5 µL 2X Kapa2G Fast Hot Start Ready Mix (Merck, Vienna, Austria), 1.25 µL of each primer (10 µM), 2 µL 25 mM MgCl_2_, 6 µL water, and 2 µL of DNA sample. The cycling program included an initial denaturation step at 95 °C for 1 min, 35 cycles of heat denaturation for 15 s at 95 °C, annealing for 15 s at 65 °C, elongation for 30 s at 72 °C, and a final elongation step for 10 min at 72 °C. To discriminate between pathogenic and non-pathogenic serovars and evaluate possible serovar switching over time, we selected BALF samples from two of the ten pigs in each farm at the five consecutive time points. The obtained 1130 bp amplicon was submitted for Sanger sequencing (Microsynth, Balgach, Switzerland). A phylogenetic tree with reference sequences of the *ompP2* gene for 15 known serovars of *G.p.* was built using the Maximum likelihood method with 1000 bootstrap replicates in the MEGA6 Software (https://www.megasoftware.net/; accessed on 29 December 2021). Kielstein et al. originally developed a scheme to identify lineages of bacteria by their extracellular polysaccharide capsule and to classify them according to their virulence based on their ability to cause disease after intraperitoneal inoculation in specific pathogen-free pigs [35]. We referred to Schuwerk et al. [36] for discrimination between pathogenic and non-pathogenic serovars. SIV positive samples were subtyped by RT-PCRs targeting the haemagglutinin (HA) and neuraminidase (NA) genes [37,38].

### 2.4. Assessment of Clinical Symptoms

Clinical symptoms of all pigs were recorded at each time point alongside BALF and serum sample collection. Weight and internal (rectal) body temperature were measured and the respiratory frequency was counted. General behaviour, posture, type of breath, dyspnoea, cough, skin and conjunctival colour, together with ocular as well as nasal discharge were evaluated according to the score system described in Appendix A.

### 2.5. Statistical Analysis

Based on the qPCR/RT-qPCR results, each sample was classified as positive or negative for each pathogen. The organism load of each pathogen in the positive samples was further evaluated. Coinfections with diverse combinations of different pathogens and the number of samples positive for each combination were documented (Appendix A). Differences in the number of positive samples and the number of pathogen combinations between different time points were assessed by the Friedman test [39] with Bonferroni correction. This test was also used to determine the differences between samples reflecting a coinfection with *P. suis* and those without *P. suis* at each time point. The average change in the pathogen loads between sampling time points was calculated by variance analysis with repeated measures using the Huynh–Feldt correction factor. Pairwise comparison of pathogens’ load per time point was conducted using a post hoc test with Bonferroni correction. Differences between the number of positive samples and the respective pathogen loads were determined by the Mann-Whitney-U test using the *P. suis* positive or negative rates as variable for grouping. The association between the presence of *P. suis* and the frequency of coinfection with other pathogens as well as the association between the frequency of coinfection with different pathogens and clinical symptoms were evaluated using Spearman’s coefficient of correlation *ρ*. Statistical analyses were carried out with the software IBM SPSS Statistics version 27 (IBM Corporation, Armonk, NY, USA).

## 3. Results

### 3.1. P. suis Load Varies over Time

The presence and quantity of *P. suis* and 12 viral and bacterial pulmonary pathogens were determined in a total of 248 BALF and 250 serum samples obtained from 50 pigs from five separated farms (10 pigs per farm) at five different life stages. Overall, 125 samples were *P. suis* positive (50.4%). Figure 1 shows the trends of the mean values of *P. suis sodA* gene copies/mL in each farm over time.

The change in the mean *P. suis* organism load in all samples over the 4-month observation period was significant (*p* = 0.001). At 1-WO, *P. suis* was present in three farms with a load around 10^2^ copies/mL. Two weeks later, the load in those farms either remained at the same level or decreased by one order of magnitude (mean 4.4 × 10^2^ copies/mL). The differences in *P. suis* load between 1-WO and 3-WO were not statistically significant. At 2-MO, *P. suis* load increased significantly (*p* = 0.004) in all five farms with a mean of 1.5 × 10^5^ copies/mL. One month later, *P. suis* continued either to increase, remained stable, or underwent a slight decrease depending on the farm (mean 8.8 × 10^4^ copies/mL). At 4-MO, a decrease was observed in all farms (mean 6.3 × 10^3^ copies/mL). The variation of copies/mL between 2-MO and 3-MO and between 3-MO and 4-MO was not statistically significant. *P. suis* copies/mL data are also provided in a data repository (https://doi.org/10.34876/p00r-9328; accessed on 29 December 2021).

### 3.2. P. suis Prevalence Increases with Time

Figure 2 shows the sum of the pathogen combinations in all investigated samples for the analysed time points. The overall prevalence of *P. suis* over the 4-month time course showed significant variations (*p* < 0.001). The fungus was detected as early as one week after birth (8/50 pigs, 16%) or starting from week 3 (13/50 pigs, 26%) with a slight increase in the prevalence, but with no statistical significance. At 2-MO and 3-MO, *P. suis* prevalences increased significantly (*p* < 0.001), with up to 85% and 87% of the animals turning positive, respectively. At 4-MO, the *P. suis* prevalence declined to 38% while infection rates with other agents increased significantly (*p* < 0.001). A total of seven animals (2.8%) was found free of the investigated pathogens, of those five were one week old and two were three weeks old. At successive life stages, no pig remained free of the investigated pulmonary pathogens at all.

### 3.3. Co-Infection Patterns Are Highly Variable

A total of 54 different co-infection patterns were noted, of which 30 involved *P. suis* (55.5%) (Appendix A). Among the latter 30 patterns, the most abundant was represented by *P. suis* + *G.p.* + *S.s.* + *Mycoplasma* spp. (*M.* spp.), which counted for 17% of all *P. suis* positive cases. The same co-infection pattern plus *B.b.* represented 13% of all cases, while with the additional infection of *P.m.* it was observed in 10% of the cases. The pattern including *P. suis* + *G.p.* + *S.s.* was present in 9% and *P. suis* + *G.p.* + *S.s.* + *B.b.* + *P.m.* + *M.* spp. in 8% of *P. suis* positive cases. All the remaining 24 co-infection patterns accounted for ≤4% each. Only one pig was infected with *P. suis* alone at weeks 1 and 3 (1.6% of *P. suis* positive samples).

Of the samples negative for *P. suis* (46.7%), 16% presented a co-infection with *G.p.* + *S.s.*, this being the most abundant type of infection encountered in this study, almost exclusively observed at weeks 1 and 3. Other relatively common infection patterns included 6% of coinfection with *G.p.* + *S.s.* + *M.* spp. and 5.6% of sole infection with *S.s*. All the remaining combinations accounted for less than 1% each. Besides *S.s.*, *P. suis* and *G.p.* were each detected as a sole infection in 0.8% of all cases. The number of pigs positive for each single bacterial species showed a significant variation over the observed course (*p* < 0.001).

Viruses were rarely detected in all serum samples, with only 3.2% being positive for SIV and 3.2% being positive for PCV2. No pig exhibited an infection with any of the three PRRSV strains.

Regarding the bacterial quantification data, only three bacterial species showed a significant change of load over time, namely *A.p.* (*p* = 0.039), *B.b.* (*p* < 0.001), and *M.hr.* (*p* = 0.046). Details of quantitative data for all pathogens are provided in a data repository (https://doi.org/10.34876/p00r-9328; accessed on 29 December 2021). Besides *P. suis,* the two most abundant pathogens in our study were *G.p.* (up to 2.5 × 10^8^ copies/mL at 3-MO) and *M.hr.* (up to 1.9 × 10^6^ copies/mL, also at 3-MO).

The frequency of *P. suis* positive and *P. suis* negative animals exhibited significant differences at certain time points, i.e., at 1-WO (*p* = 0.046), 2-MO (*p* = 0.046), and 3-MO (*p* = 0.025). Coinfection with *G.p.* + *S.s*. occurred significantly more often than coinfection with *G.p.* + *S.s.* + *P. suis* (*p* = 0.035). A total of 13 co-infection patterns involved *P. suis*, while seven did not. At month 3, the total number of bacterial co-infections involving *P. suis* was significantly higher when compared to those not involving the fungus (*p* = 0.035). When considering the bacterial loads of each single species, *M.hr.* and *B.b*. were present at month 2 in a significantly different amount when together with *P. suis* than without. In particular, *M.hr.* was significantly higher in the first case, while *B.b*. was lower. Figure 3 shows the trend of the same three classes (*P. suis* positive pigs, *P. suis* negative pigs, and pigs with no pathogen detected) over time for each separate farm.

*P. suis* was detected in pigs from only three out of five farms at week 1, while in the remaining two farms from week 3 onwards. The infection rate peaked at month 2, with *P. suis* being detected in 43 out of 49 sampled pigs (86%). Only two and five pigs in two farms remained negative. The infection rate remained high at month 3, with *P. suis* being detected in either 9 out of 10 pigs in two farms, in all 10 pigs in other two farms, or in 50% of the animals in the fifth farm. In all farms, we observed a substantial decline of the infection rate at month 4, with only three to six *P. suis*-positive pigs per farm.

### 3.4. P. suis Co-Infection Scenario Varies over Time

The pie chart of Figure 4 shows the prevalence of different patterns of co-infection with *P. suis* and other pathogens grouped by the number of co-infecting bacterial or viral species detected. The highest prevalence was recorded for the combination of *P. suis* and four other pathogens, detected in 30.4% of pigs, followed by the combination of *P. suis* and three others (25.6%), *P. suis* and five others (19.2%), *P. suis* and two others (14.4%), *P. suis* and one other (4.8%), *P. suis* and six others (4%), and *P. suis* single infection (2%).

The relative prevalence of the seven recorded co-infection patterns was significantly different over time (*p* = 0.004; Figure 4). In the first three weeks, only co-infections of *P. suis* with a maximum of three other pathogens were observed, while in the following life stages the complexity of the co-infections increased. At months 2 and 3, co-infections of *P. suis* with three and four other pathogens were most abundant and occurred in 15.3% and 8% of the cases, respectively. One month later, co-infections of *P. suis* with four and five other pathogens dominated and could be detected in 17.7% and 10.5% of the cases, respectively. At month 4, the prevalence of different co-infection patterns became similar (3.5% each), except for a relatively low prevalence of co-infection of *P. suis* with six other pathogens detected in only 1.6% of cases at this stage.

Statistical analysis of all co-infection patterns revealed that *P. suis* had a significant association with the combinations *G.p.* + *M.* spp. + *S.s.* (*p* = 0.030) and *B.b.* + *M.* spp. + *P.m.* + *S.s.* (*p* = 0.045). The analysis of possible associations between *P. suis* and the various pathogen combinations revealed differences between the farms (Appendix A).

### 3.5. P. suis-Specific Clinical Signs Could Not Be Detected

None of the co-infection combinations involving *P. suis* showed a correlation with any of the evaluated clinical symptoms. Instead, *G.p.* + *S.s.* co-infection correlated with an alteration of breath type (*ρ* = 0.43, *p* = 0.032) and *G.p.* + *S.s.* + *P.m.* with the occurrence of ocular discharge (*ρ* = 0.40, *p* = 0.044). When samples were sorted out by farms, more co-infections showed a correlation with clinical signs. In Farm 2, coinfection with *P. suis* + *G.p.* + *M.* spp. + *S.s.* correlated with altered respiration frequency. In Farm 5, single infection with *P. suis* correlated with dyspnoea (*ρ* = 0.889, *p* = 0.044), coinfection with *P. suis* + *G.p.* + *P.m.* + *S.s.* exhibited a correlation with nasal discharge (*ρ* = 0.88, *p* = 0.047), while single infection of *S.s.* showed a correlation only with weight (*ρ* = 0.89, *p* = 0.041). When sorting the samples by time, many more co-infections showed a correlation with respiratory symptoms, especially with the presence of cough and nasal discharge (see Appendix A for more details). In general, none of the clinical signs was associated with a specific pathogen combination including *P. suis* or the same combination without *P. suis*. The detailed record of the clinical symptoms in the sampled pigs over time is given in a repository file (https://doi.org/10.34876/yf4n-6902; accessed on 29 December 2021).

### 3.6. G. parasuis Serotyping and SIV Subtyping

Longitudinal analysis of the *G.p.* serovars of two pigs per time point showed a switch from non-pathogenic to pathogenic serovars starting from month 2, with the number of pigs tested positive for various pulmonary pathogens increasing significantly. A total of eight serum samples were positive for SIV. Unfortunately, a subsequent sequencing of the HA and NA genes was not successful, most likely due to the low viral loads in these samples.

## 4. Discussion

*Pneumocystis* spp. are a group of opportunistic fungi responsible for severe pneumonia in immunocompromised individuals. Although its presence in the pig has long been known since the 1950s, the role of *P. suis* in porcine pulmonary disease has been mostly neglected.

The present longitudinal study is, to our knowledge, the first to quantitatively analyse *P. suis* and 14 other common viral and bacterial respiratory pathogens in conventional farm-raised pigs and evaluate their correlation with clinical symptoms. Despite an uneven prevalence of *P. suis* among the five investigated farms, this fungus was present as early as in the suckling piglets. The presence of *Pneumocystis* spp. in early life stages has already been reported in this host [14,16,17,29], as well as in humans and other mammals [40,41,42,43]. While the fungal load was very low at these early stages (between 10^2^ and 10^3^ copies/mL), it increased significantly later on, reaching up to 10^5^ genome copies/mL between the third week and the second/third month. At the same life stages (weaned piglets), we also observed the highest prevalence of *P. suis* infection. Of note, despite a significant decrease in the *P. suis* infection rate in fattening pigs at the fourth month of life, the *P. suis* organism load remained comparable to that at the earlier stages. These results appear to be in line with a previous study, which reported the highest prevalence of *P. suis* in the post-weaning phase [26] or the highest organism load in two-month-old piglets as determined by qPCR [29]. However, another study indicated that the prevalence of *P. suis* was higher in suckling piglets than in weaners or fatteners [21]. This might be explained by the co-infection with PRRSV in those piglets, given the immunosuppressive effect of this virus [44].

In light of the findings that *Pneumocystis* spp. has a significantly reduced genome size and lacks many crucial biosynthetic pathways required to survive outside of the host [4], it is currently accepted that this pathogen is an obligate biotroph [45,46,47,48,49], depending on the host lungs for nutrients. Yet, this study demonstrated that *P. suis* is able to cope with several co-infecting agents at the same time, even though this might not be valid for an extended period of time, thus explaining its decreased prevalence at four months.

Co-infections of *P. suis* with viruses and bacteria have been reported in a few previous studies of farmed pigs. Kim et al. observed a high prevalence of *P. suis* positive pigs co-infected with both PCV2 and PRRSV (48.7%), or with only PRRSV (12.8%) [24]. In our earlier retrospective study, the most frequent co-infection involving *P. suis* was that including PCV2 [14]. PRRSV and PCV2 are known for acting as primary pulmonary pathogens in the porcine host [50], causing initial infections and/or suppression of the immune system allowing secondary pathogens to invade the host, often leading to a worsened pathological outcome. With regard to *P. suis*, it is currently hypothesized that PRRSV and PCV2 play a role in increasing the chance of an infection with the fungus, as Kim et al. observed a higher percentage of *P. suis* positive samples in those positive for both viruses than in those positive for only one virus (23% vs. 14.3%) [26]. Unexpectedly, in the current study none of the PRRSV strains under investigation was detected and only a small percentage of samples was SIV or PCV2 positive, which accounted for the latter only at the fourth month of age. Our results suggest that the presence of PRRSV and PCV2 is not a precondition for *P. suis* to colonize and proliferate in the porcine host. This is consistent with our previous study, in which all *P. suis* positive pigs were negative for PCV2, while PRRSV was present only in suckling piglets [29]. Some bacterial species are also primary respiratory pathogens in pigs and have been reported to be able to impair the host immune system. Among those, *Mycoplasma* spp. are for instance responsible for the alteration of cytokine responses and macrophage function [50]. Synergistic effects are also known to occur, as some pathogen combinations lead to a more severe disease outcome [50,51].

In the present study, we observed very infrequent viral infections, but widespread bacterial infections, which were present at every time point, with the number of concomitant bacterial species increasing over time to up to six. The most prevalent co-infections were those comprising three or four bacterial species. Among those, co-infections with *G. parasuis* and *S. suis* were the most common, present alone or in combination with *Mycoplasma* spp., *Bordetella* spp., or *P. multocida*. *G. parasuis* was the most frequently detected species in this study regardless of the time point, and present at high loads as early as the first week of life. Given its known role as a primary pathogen and the fact that we observed a switch from non-pathogenic to pathogenic serovars over time, it might contribute to the proliferation of *P. suis*. We did not observe any association of clinical symptoms with specific co-infections involving *P. suis*, which is not surprising, given the very complex pathophysiologies of multiple co-infections. Based on our experiences of routine diagnostics of porcine pneumonia by *in situ* hybridization, *P. suis* is rarely present at high loads in the alveoli of pigs. In lung tissues affected by different types of pneumonia, *P. suis* signals are usually restricted to small areas with minor lesions [14]. These observations may suggest poor growth and proliferation of *P. suis* in affected lungs due to suboptimal conditions, such as poor oxygen availability or lack of sufficient nutrients resulting from tissue damage caused by abundant and prolonged bacterial infections [14]. The presence of only minor lesions in the infected lungs may reflect the inhibitory effects of *Pneumocystis* trophic forms on inflammatory processes caused by the β-glucan released from *Pneumocystis* cysts as suggested from a recent study in a mouse model [52].

In contrast to previous observations [14], the present study exhibited a high *P. suis* load coexisting with several other pathogens, suggesting an active proliferation despite the presence of multiple pathogens. However, it was not possible to address the spatial distribution of *P. suis* and other co-infecting pathogens as we used BALF instead of lung tissue samples. The current study is limited to 14 commonly known pulmonary pathogens. A more thorough analysis would have been obtained by high-throughput DNA sequencing such as metagenomic next-generation sequencing and targeted rRNA sequencing of the 16S or internal transcribed spacer regions.

In addition, predisposing conditions for *Pneumocystis* colonization and proliferation should be unravelled. The current study is lacking specific immune response assessment in BALF, sera, and colostrum. In humans, as well as in rodent models, it is well known that immunosuppression is crucial for pathogenesis. This might also be the case in the pig, either due to the immunosuppressive impact of some pathogens, environmental and farm conditions, or inherited immunodeficiency diseases of certain pig breeds [53,54,55]. Post-weaning immunosuppression, which can be caused by nutritional changes, decline of maternal antibodies, as well as social stress factors due to mixing of weaners, is also known to pave the way for PRDC [56,57]. Additionally, the piglet’s immune system is not fully developed after birth, making the animals susceptible to pulmonary infection [58]. All these factors will likely contribute to the flare-up of pathogenic agents during this critical period. Regarding the decline in *P. suis* prevalence in all the farms under investigation at the fourth month of life, we hypothesize that this might be due to the clearance of *P. suis* organisms from the lungs as the immune system increasingly matures, or a lower ability of *P. suis* to compete with other pathogens for space and nutrients in the lungs as evidenced by the high prevalence of infections with bacteria detected at this time point.

## 5. Conclusions

Through a longitudinal quantitative assessment of the microbial community in the lungs of pigs from multiple farms along with respiratory symptoms, we identified complex co-infection patterns involving *P. suis* and other pulmonary pathogens in pigs, suggesting a potentially relevant role of *P. suis* in the pathogenesis of PRDC rather than representing a subclinical infection in this host. Further research is warranted to better understand this complex interaction.

## Figures and Tables

**Figure 1 jof-08-00043-f001:**
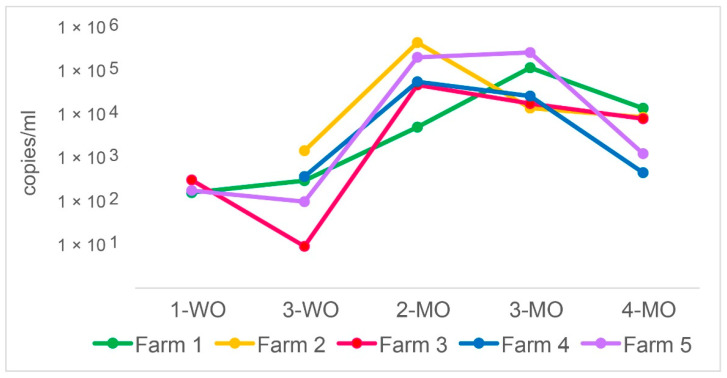
Dynamics of the *P. suis* organism loads at different life stages. The *Y* axis represents the *P. suis* organism loads expressed by *sodA* gene copies/mL as determined by qPCR. The *X* axis shows the 5 life stages from the first week (1-WO) to the fourth month (4-MO) of life.

**Figure 2 jof-08-00043-f002:**
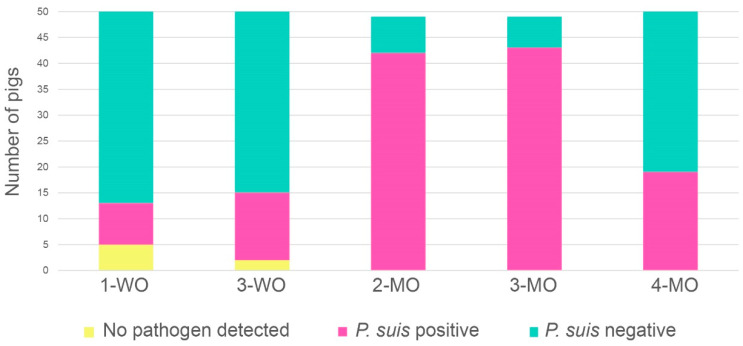
Overall *P. suis* prevalence over time. *P. suis* positive pigs (*P. suis* single-infections and co-infections with *P. suis* and other pathogens) are shown in pink, *P. suis* negative pigs (positive for other pathogens) are shown in turquoise and pigs without any pathogen detected are shown in yellow.

**Figure 3 jof-08-00043-f003:**
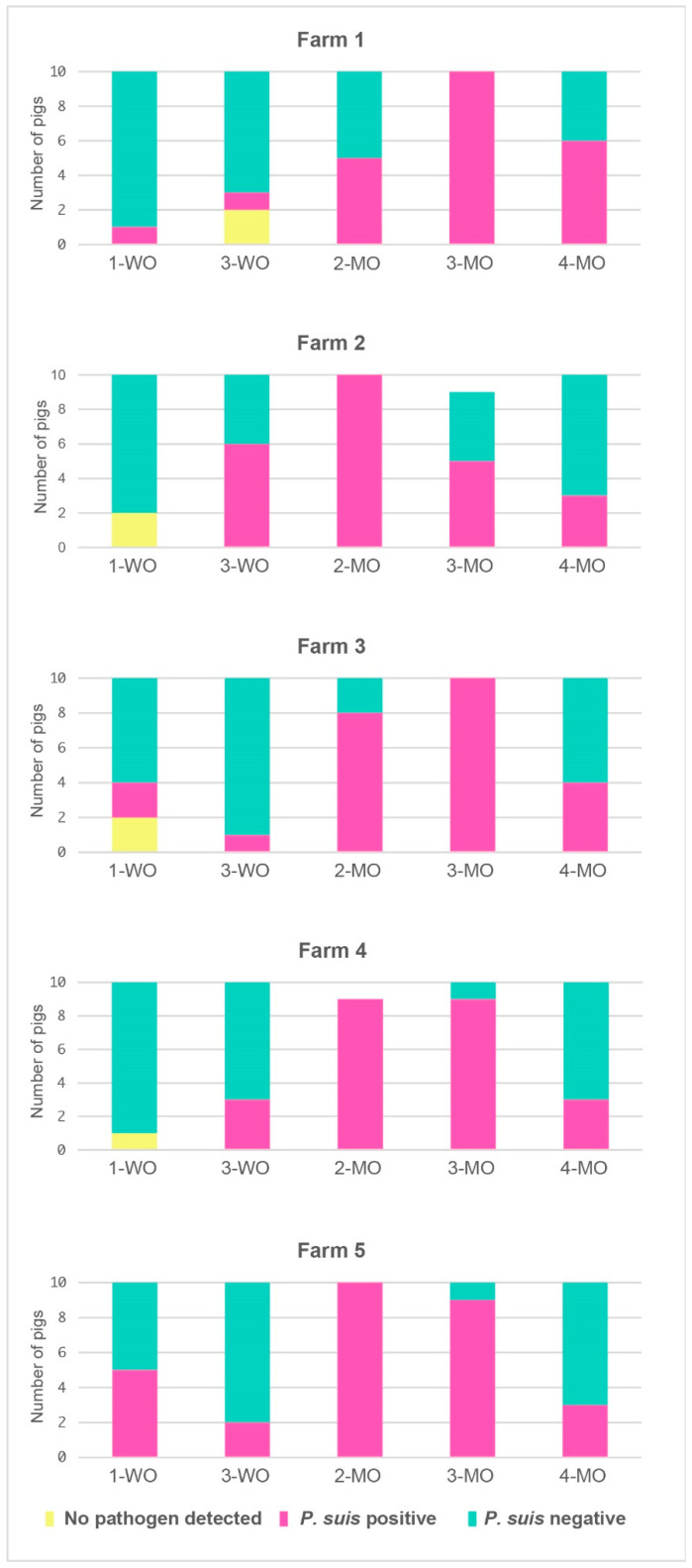
Prevalences of *P. suis* over time at farm level.

**Figure 4 jof-08-00043-f004:**
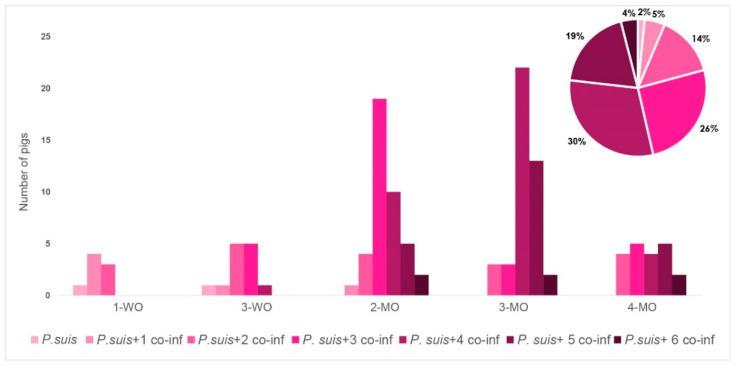
Prevalence of different *P. suis* co-infection combinations over time.

**Table 1 jof-08-00043-t001:** Primers and probes used in the current study.

Species	Gene	Forward primer 5′-3′	Reverse Primer 5′-3′	Probe 5′-3′	Annealing Temperature	Reference
*P. suis*	*sodA1*	GGCGAGTTAGCTGCAATTCAA	AGCACATTATAGGACCATTGTTGT	FAM-ACAAGCCAACACCACCCACTTCC-TAMRA	55 °C	This study
	*sodA2*	GGCAGTGTAAATCAATTTAT	AAACGAGTCAAATAAATACA	FAM-AGGAAGTGGATGGTGTTGGC-TAMRA	55 °C	
PCV2	capsid protein	GGTACTCCTCAACTGCTGTCC	GGGAAAGGGTGACGAACTGG	FAM-ACAGAACAATCCACGGAGGAAGGG-TAMRA	60 °C	Weissenbacher-Lang 2017 [29]
PRRSV-EU	N (nucleocapsid protein)	GCACCACCTCACCCRRAC	CAGTTCCTGCRCCYTGAT	FAM-CCTCTGYYTGCAATCGATCCAGAC-BHQ1	55 °C	Wernike et al., 2012 [30]* This study
PRRSV-US	N (nucleocapsid protein)	ATRATGRGCTGGCATTC	ACACGGTCGCCCTAATTG	JOE-TGTGGTGAATGGCACTGATTGACA-BHQ1 *	55 °C	
PRRSV-HP	*pp1a*	CCGCGTAGAACTGTGACAAC	TCCAGGATGCCCATGTTCTG	CY5-ACGCACCAGGATGAGCCTCTGGAT-TAMRA	55 °C	
SIV	matrix protein M	AGATGAGTCYTCTAACCGAGGTCG	TGCAAARACAYYTTCMAGTCTCTG	FAM-TCAGGCCCCCTCAAAGCCGA-TAMRA	60 °C	Bonin et al., 2018 [31]
*A. pleuropneumoniae*	*apxIVA*	GTGGTTTGGAAAGTATTATC	TTTAAACCTTGTTTCGTCTA	FAM-CGGTATCGGTGGAACGGTAA-TAMRA	55 °C	This study
*G. parasuis*	*infB*	CGACTTACTTGAAGCCATTCTTCTT	CCGCTTGCCATACCCTCTT	FAM-GCACTTAATTCTAATACTTCCGAT-TAMRA	55 °C	Turni et al., 2010 [32]
*M. flocculare*	*fruA*	TTAGCAGTTCCAATTTTATCAG	AAACCATAGGTATCTTTAAGTTG	FAM-CAATTCCGCCAACTACAAATCCAG-BHQ1	55 °C	Fourour et al., 2018 [33]* This study
*M. hyopneumoniae*	P102	TAAGGGTCAAAGTCAAAGTC	AAATTAAAAGCTGTTCAAATGC	CY5-AACCAGTTTCCACTTCATCGCC-BHQ2	55 °C	
*M. hyorhinis*	P37	TTCTATTTTCATCTATATTTTCGC	TCATTGACCTTGACTAACTG	CY3-CAGGAGTAGTCAAGCAAGAGGATG-BHQ2 *	55 °C	
*P. multocida*	*sodA*	GGAAGCCTTCCAAGCAGAATTTG	CCGCAATAGCTTTACCCATTACAG	CY5a-CGGCAGCAACCCGTTTCGGTTCAG-BHQ2b	60 °C	Tocqueville et al., 2017 [34]
*S. suis*	*rsmG*	TGAAGCATTTCTACGATTCT	GCGGAAAGATAATCTTCATA	FAM-TTGGAGCTGGAGCTGGATTC-TAMRA	52 °C	This study

* The probe was designed for this study and used with the previously published primers.

## Data Availability

Publicly available datasets were analyzed in this study. This data can be found here: https://doi.org/10.34876/yf4n-6902 (accessed on 29 December 2021) and https://doi.org/10.34876/p00r-9328 (accessed on 29 December 2021).

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
