# Peer review of "Pneumocystis spp. in Pigs: A Longitudinal Quantitative Study and Co-Infection Assessment in Austrian Farms"

_jof, 2021, doi:10.3390/jof8010043_

Round 1

Reviewer 1 Report

The manuscript from Blasi, et al. is a longitudinal study of Pneumocystis suis infection in piglets on 5 different Austrian farms.  The authors also examined co-infection with a number of common bacteria and viruses that infect swine.  The data is descriptive and fails to incorporate any measures of immunity from the bronchial alveolar lavage or serum samples collected over time.  This seems to be a lost opportunity as the authors speculate in the discussion that their findings of infection in the animals could be due to the naïve immune system.  The patterns of infection across the farms vary and it is difficult to discern from the data any important relationships.  Overall, the authors conclude that Pneumocystis contributes to a complex coinfection history and suggest the fungus plays a relevant role in the respiratory diseases of the piglets.  The manuscript could be strengthened by examining antibody responses to Pneumocystis in the BAL and perhaps in the mothers milk if that was collected.  There is plenty of evidence in murine models that Pneumocystis is harbored in the lungs of infant mice till weaning when an adaptive immune response is mounted.  There is also new data from murine experiments suggesting that the trophic forms of Pneumocystis inhibit inflammation driven by the beta glucans of the ascus forms.  Perhaps this should be discussed in the context of the interactions between the co-infections.

Author Response

Dear Reviewer 1,

We are thankful for the constructive comments and critics to our manuscript.

We agree with you that the aspect of the immune response is of interest and value to this work in addition to the current results. Unfortunately, this was not included in our original experimental set up. We have processed all BALF samples by centrifugation and subsequent disposal of supernatant. Most of the cell pellets have been used for extensive molecular analysis as described in the manuscript, with only a tiny amount of leftover available for some samples. In addition, we didn’t collect colostrum. Therefore, at this point we’re unable to assess the antibody responses in BALF or milk. We have included this information as a limitation of this study in the Discussion section (page 14, lines 448-449).

We have integrated the discussion with the literature regarding the harbouring of the fungus in the life stages up to the end of the weaning (page 12, lines 370-372).

As suggested, we have also discussed the role of Pneumocystis trophic forms in inflammation inhabitation in the revised manuscript (page 14, lines 435-438).

Reviewer 2 Report

This is a very original and highly interesting study. It is an excellent longitudinal study with perfect analyses and stats. Moreover, its subject is highly relevant economically. I have only the following comments that are only meant to improve the paper.

Major comments

  1. The analysis is limited to the 12 pathogens that were looked for using specific qPCR. This limitation could be mentioned in the paper. In a subsequent study, whole genome sequencing could be used in order to obtain a complete view. 
  2. line 285 : is the difference at 3-wo really not significant ?
  3. line 306: could acquisition of immunity against P suis have played a role or be postulated ?
  4. the quality of the images of figures 1, 2 and 3 should be improved for final publication. That of figure 4 is fine.

Minor comments

  1. line 61: it should be explained how it varies.
  2. line 216: it should be mentioned that the pigs were "standard farm-raised".
  3. line 227 and line 249: the test used should be mentioned.
  4. line 236: the access to the suppl data on https://phaidra.vetmeduni.ac.at/o:845 should be improved.
  5. line 239-241: the decription of the groups should be deleted in the text because it is a duplicata of that in the legend of the figure, where it is necessary.
  6. Table S3: p and p should be explained in the footnote.
  7. line 379: other relevant references should be cited.
  8. lines 450-451: i could not access to the zenodo page, the link might be wrong.

Author Response

Dear Reviewer 2,

we are thankful for your appreciation of the manuscript and for your insightful comments and suggestions, which are very helpful to improve the manuscript. Please find our point by point responses below.

Major comments and answers:

1. The analysis is limited to the 12 pathogens that were looked for using specific qPCR. This limitation could be mentioned in the paper. In a subsequent study, whole genome sequencing could be used in order to obtain a complete view.

1A. Thanks for pointing out the limitation of this study. We totally agree with it. Indeed, we considered the possibility of metagenomic and targeted 16S/ITS NGS. However, we encountered a number of issues with the quantity and quality of the BAL samples. Most of the BAL samples have been used for extensive PCR testing as described in the manuscript. There is only a tiny amount of leftover available for a small number of samples, which prevents us doing metagenomic or targeted 16S/ITS analysis with adequate statistical power or meaningful interpretation. We have discussed this limitation in the revised manuscript (page 14, lines 443-447).

2. line 285 : is the difference at 3-wo really not significant ?

2A. We checked the significance value of the difference between the class of P. suis infected and non-infected pigs and can confirm that there is no significance (p = 0.18).

3. line 306: could acquisition of immunity against P suis have played a role or be postulated ?

3A. The acquisition of immunity against P. suis has highly likely played a role. We have added this aspect in the discussion (page 15, lines 460-462).

4. The quality of the images of figures 1, 2 and 3 should be improved for final publication. That of figure 4 is fine.

4A. We agree that the image quality of figures 1, 2 and 3 needs improvement. Upon submission of the manuscript, we uploaded each figure in .pdf format and in high resolution. We confide that those will be used for the formatting of the final manuscript in case of acceptance for publication.

Minor comments and answers:

1. line 61: it should be explained how it varies.

1A. We adjusted the text accordingly and made an update of the literature.

2. line 216: it should be mentioned that the pigs were "standard farm-raised".

2A. A detailed description of the farms in present in the Material and Methods section and we didn’t want to repeat this information in the Results section. Both in the Materials & Methods and in the discussion (lines 103 and 367, respectively) we also define the pigs included in this study as conventional farm raised. 

3. line 227 and line 249: the test used should be mentioned.

3A. The tests adopted for the results mentioned in line 227 and 249 are now listed in more details in the Materials and Methods section (paragraph 2.6 Statistical analysis). In the results section we would prefer not to repeat this information in order not to affect the fluidity of the text for the reader.

4. line 236: the access to the suppl data on https://phaidra.vetmeduni.ac.at/o:845 should be improved.

4A. Our University just provided us the corresponding DOIs for the data repository. The text has been updated accordingly and the readers will have access to the files once clicked on the provided link using the download function on the right side of the page.

5. line 239-241: the description of the groups should be deleted in the text because it is a duplicata of that in the legend of the figure, where it is necessary.

5A. We have deleted the duplicated text from the main text and kept it only in the Figure caption.

6. Table S3: p and p should be explained in the footnote.

6A. We have added the meaning of p and p in the footnote of the Supplementary Table S3 and S4.

7. line 379: other relevant references should be cited.

7A. We improved the citation of references (page 13, lines 386-389).

8. lines 450-451: i could not access to the zenodo page, the link might be wrong.

8A. The correct link has been now updated in the manuscript (page 15, line 471).

Round 2

Reviewer 1 Report

The authors addressed previous comments by adding some commentary to the discussion regarding weaknesses of the study and another potential interpretation of the data.  The authors didn't mention in the methods section how the 10 piglets from each farm was chosen for the study.  Were they randomly chosen or were they chosen because they had symptoms of respiratory infection.  This is important since the authors indicate the bacteria found in the lungs were infections and not commensal.  

Author Response

Dear Reviewer 1,

once again we are thankful for the constructive comments and the accurate review.

We improved the language of the manuscript.

We agree that the method of selection of the animals plays a role. The farms were chosen according to previous histories of respiratory diseases and also on farm level, piglets with respiratory symptoms were selected for sampling. This aspect was included in various parts of the manuscript (lines 28, 102-103, 106-107, 456).